# Hypoxia-Induced Extracellular Vesicles and Non-Coding RNAs in Cancer: A Systematic Review of Tumor Dynamics and Therapeutic Implications in Preclinical Animal Models

**DOI:** 10.3390/biomedicines13112796

**Published:** 2025-11-17

**Authors:** Joao Pedro R. Afonso, Simona Taverna, Annalisa Pinsino, Giuseppe Cammarata, Rodrigo A. C. Andraus, Iranse O. Silva, Carlos H. M. Silva, Claudia S. Oliveira, Rodrigo F. Oliveira, Deise A. A. P. Oliveira, Orlando A. Guedes, Luciana P. Maia, Wilson R. Freitas Junior, Elias J. Ilias, Juan J. Uriarte, Giuseppe Insalaco, Luis V. F. Oliveira

**Affiliations:** 1Graduate Program in Human Movement and Rehabilitation (PPGMHR), Evangelical University of Goiás (UniEVANGELICA), Anápolis 75083-515, GO, Brazil; joaopedro180599@gmail.com (J.P.R.A.); rodrigoandraus@gmail.com (R.A.C.A.); iranseoliveira@hotmail.com (I.O.S.); carloshmendes@unievangelica.edu.br (C.H.M.S.); csantos.neuro@gmail.com (C.S.O.); rodrigofranco65@gmail.com (R.F.O.); deisepyres@gmail.com (D.A.A.P.O.); orlandoaguedes@gmail.com (O.A.G.); lucianapmaia@gmail.com (L.P.M.); 2Institute of Translational Pharmacology, National Research Council of Italy (CNR), 90146 Palermo, Italy; simona.taverna@cnr.it (S.T.); annalisa.pinsino@cnr.it (A.P.); giuseppe.cammarata@cnr.it (G.C.); giuseppe.insalaco@ift.cnr.it (G.I.); 3Graduate Program in Health Sciences (PPGCS), Faculty of Medical Sciences of Santa Casa de São Paulo, São Paulo 01224-001, SP, Brazil; wilsonrfreitasjunior@gmail.com (W.R.F.J.); eliasjilias@gmail.com (E.J.I.); 4Nebrija Research Group ARIES, Higher Polytechnic School, Antonio de Nebrija University, 28015 Madrid, Spain; juriarte@nebrija.es

**Keywords:** hypoxia, animal models, extracellular vesicles, non-coding RNAs, tumor progression

## Abstract

**Background:** Cancer is a heterogeneous pathology, and among causative factors, gene expression can influence its development. Molecular approaches using extracellular vesicles (EVs) and non-coding RNAs (ncRNAs) offer great value in understanding tumor progression, early diagnosis, and potential therapies. **Objectives:** This systematic review was conducted in accordance with the Preferred Items for Reporting of Systematic Reviews and Meta-Analyses guidelines. Its main objective was to evaluate the effects of cellular hypoxia in different types of cancer, exclusively using animal models and highlighting the regulatory role of microRNAs and circular RNAs in tumor development. **Methods:** A literature review was performed using the PubMed/Medline and Scopus databases without year limitations. The initial search yielded 171 articles. After applying the inclusion and exclusion criteria, 25 studies were included in this review. Data analysis showed that animal models provide detailed insights into different types of cancers under hypoxic conditions. **Results:** Our analysis identified that specific circRNAs, such as circPFKFB4 in breast cancer and circPDK1 in pancreatic cancer, are consistently associated with a worse prognosis and therapeutic resistance. Similarly, miRNAs such as miR-1287-5p (breast cancer) and miR-133a (colorectal cancer) have frequently been identified as tumor suppressors whose levels are altered by hypoxic conditions. Furthermore, the results suggested that in some cancers, the release of EVs may facilitate tumor progression and metastasis. However, manipulation of ncRNA expression causes significant changes in the tumor response, which suggests a therapeutic response. **Conclusions:** This study shows that the use of animal models is essential for exploring the molecular mechanisms of cancer and establishing new therapeutic approaches.

## 1. Introduction

Cancer is a heterogeneous pathological condition arising from various factors, including dysregulated gene expression and alterations in the tumor microenvironment (TME) [1]. Research shows that approximately 3% of coding sequences of human DNA are translated into protein, but over two-thirds (75%) are transcribed into non-coding RNAs (ncRNAs) [2]. Over the years, various types of non-coding RNAs (ncRNAs) have been extensively studied, including microRNAs (miRNAs), long non-coding RNAs (lncRNAs), circular RNAs (circRNAs), and Piwi-interacting RNAs (piRNAs), with each playing distinct roles, predominantly in gene expression regulation [3,4,5].

Among the various types are miRNAs, which are small RNA molecules capable of regulating gene expression post-transcriptionally. They function by binding to precise regions of target messenger RNAs (mRNAs), leading to their degradation or the repression of translation [4]. On the other hand, circRNAs are notable for their high stability and can act in various ways. However, their most prominent role is their ability to act as “miRNA sponges”; they sequester miRNAs, preventing them from acting on their respective target mRNAs and thus causing an indirect regulation of gene expression [4,5].

Advances in high-throughput and high-precision technologies, such as genome sequencing and bioinformatics, have greatly improved our understanding of cancer, its biological development, and treatment [6]. These ncRNAs are involved in multiple cellular activities, including target RNA degradation, inhibition of transcription factors, and regulation of protein interactions, thereby contributing to the development of many cancers [7]. Among the many classes of ncRNAs, miRNAs have gained critical relevance in cancer biology due to their ability to target and inhibit gene expression, which can help in both cancer progression and regression [8]. Conversely, cancer ncRNAs like lncRNAs and circRNAs are important in carcinogenesis because they can regulate gene expression at the transcriptional level and protein interactions through their binding to DNA, RNA, or proteins [4]. Hence, ncRNAs are central to temporal changes in gene expression and can serve as a basis for designing treatment regimens for neoplasms [9].

Oxygen deficiency, or hypoxia, is also a typical characteristic of the vast majority of solid tumors with an approximate prevalence of 90% [10]. The hypoxic TME activates hypoxia-inducible transcription factors (HIFs) that alter the expression of hundreds of genes and facilitate the metabolic reprogramming of the cell for low oxygen conditions [11]. HIFs are critical for cellular transcriptional reprogramming especially in processes such as angiogenesis; they also promote tumor growth and resistance to therapies [12,13,14,15]. These alterations in tumor adaptation to an oxygen-deficient environment create obstacles in cancer management; therefore, studying HIFs in cancer biology is a highly relevant and important issue [16].

Extracellular vesicles (EVs) have emerged as an important research area. These are a class of cell-derived nanoparticles that are released from cells and are involved in cell–cell communication as well as many functions in both health and disease, including cancer [17,18,19].

The term “extracellular vesicles” (EVs) refers to a diverse population of particles. Although they are sometimes classified simply by size (such as “small EVs” < 200 nm and “large EVs” > 200 nm), the main types of EVs are differentiated by their biogenesis [20]. Exosomes are generated intracellularly through inward budding of the endosomal membrane, forming multivesicular endosomes (MVEs), which release their contents upon fusing with the plasma membrane [20]. In contrast, microvesicles are formed by the direct outward budding of the plasma membrane. A third type, apoptotic bodies, are fragments released from cells undergoing apoptosis [20,21].

EVs can also alter the microenvironment of the target cells through bioactive molecules such as ncRNAs, miRNAs, and circRNAs, which are of great relevance in early cancer diagnosis and surveillance [22]. In addition to their low toxicity and biocompatibility, the ability of EVs to accumulate at physiological sites makes them ideal candidates for cancer research and therapeutic development [23,24]. To develop new therapeutic approaches, it is critical to examine the relationship between ncRNAs and gene regulation under hypoxic conditions. Whether inflammation is present or not, the induction of ncRNAs in tumors has been increasing rapidly, highlighting the potential for ncRNA-based treatments for cancers both with and without non-tumor hypoxic environments [25,26].

Due to the heterogeneity of the factors that lead to cancer development, it is necessary to develop study models to understand the molecular mechanisms of these processes. Although in vitro models offer valuable insights for a better understanding of the disease, they still have limitations that prevent the accurate assessment of their effects on the organism. Therefore, animal models should be used as a valuable method to observe the dynamics of factors that lead to the development of cancers in living organisms, enabling the reproduction of complex cellular interactions and validation of biomarkers and therapeutic targets.

Therefore, this review aimed to evaluate the effects of cellular hypoxia in different types of cancer, exclusively in animal models, emphasizing the role of miRNAs and circRNAs in tumor development and their functions as possible biomarkers and therapeutic targets. The analysis of animal models allows us to identify molecular patterns related to gene expression and provides insights that in vitro models cannot provide, confirming the importance of this study for a better understanding of cancer under hypoxic conditions through translational research.

## 2. Material and Methods

This systematic review was conducted in accordance with the Preferred Items for Reporting of Systematic Reviews and Meta-Analyses (PRISMA) guidelines (Appendix A) [27]. The protocol for this review is registered in PROSPERO under code CRD420251151574.

### 2.1. Eligibility Criteria

Studies were selected that utilized animal models to investigate the role of ncRNAs (such as miRNAs and circRNAs) in cancer progression under hypoxic conditions. Within this scope, studies that also investigated the role of extracellular vesicles (EVs) as carriers or mediators of these ncRNAs were identified and analyzed as a subset of interest. These studies compared hypoxic and normoxic conditions to explore the role of ncRNAs in tumor progression and cellular responses to hypoxic stress. Review articles, in vitro studies that did not involve animal testing, case reports, case series, opinions, or abstracts were excluded.

### 2.2. Search Information

An electronic literature search was conducted in PubMed/Medline and Scopus without limitations on the year of publication in February 2025. In addition, a manual search of the references of the studies and experts was conducted to identify publications that were not identified electronically.

### 2.3. Search Algorithms

The following keywords and their combinations were used:(“Hypoxia”[Mesh] OR Oxygen Deficiencies OR Oxygen Deficiency OR Anoxemia) AND (“Neoplasms”[Mesh] OR Tumors OR Neoplasia OR Neoplasias OR Neoplasm OR Tumor OR Cancer OR Cancers OR Malignant Neoplasm OR Malignancy OR Malignancies OR Malignant Neoplasms OR Benign Neoplasms OR Benign Neoplasm) AND (“RNA, Circular”[Mesh] OR circRNA OR circRNAs OR Circular RNA OR Circular RNAs OR Closed Circular RNA OR Circular Intronic RNA OR ciRNA).(“Neoplasms”[Mesh] OR Tumors OR Neoplasia OR Neoplasias OR Neoplasm OR Tumor OR Cancer OR Cancers OR Malignant Neoplasm OR Malignancy OR Malignancies OR Malignant Neoplasms OR Benign Neoplasms OR Benign Neoplasm) AND (“Hypoxia”[Mesh] OR Oxygen Deficiencies OR Oxygen Deficiency OR Anoxemia) AND (“Extracellular Vesicles”[Mesh] OR Extracellular Vesicle OR Exovesicles OR Exovesicle OR Apoptotic Bodies OR Apoptotic Body) AND (“MicroRNAs”[Mesh] OR Micro RNA OR MicroRNA OR miRNA OR miRNAs OR Small Temporal RNA OR stRNA OR Primary MicroRNA OR pri-miRNA OR pri miRNA OR Primary miRNA OR pre-miRNA OR pre miRNA) AND (“RNA, Circular”[Mesh] OR circRNA OR circRNAs OR Circular RNA OR Circular RNAs OR Closed Circular RNA OR Circular Intronic RNA OR ciRNA).

### 2.4. Study Selection

The researchers conducted this review using a two-stage process, as illustrated in Figure 1. In phase one of the JPRA and ST, two authors independently evaluated the titles and abstracts to identify studies that fulfilled the eligibility criteria. The researchers included studies that met the inclusion criteria and adhered to strict guidelines for further review. In the second phase, the authors independently checked the full texts of the selected studies to confirm their inclusion. A third author GC resolved disagreements when the initial reviewers could not reach an agreement but only after they had discussed the issues as necessary.

### 2.5. Data Extraction

The same independent double review process was used to gather all relevant data and facilitate subsequent comparisons. Any disagreements encountered during this phase were addressed through discussion and, when necessary, a final consensus was reached with the assistance of a third author.

The same independent double review process was used to collect all relevant data and facilitate subsequent comparisons. Only experimental studies were included with a specific focus on in vivo studies involving animals and human-derived cells or tissues implanted in xenotransplanted animal models. Any disagreements encountered during this phase were resolved through discussion and, when necessary, a final consensus was reached with the assistance of a third reviewer. Descriptive characteristics of all studies included in the analysis were extracted, including details such as author names, year of publication, model type (in vivo), and specificities of in vivo models (including animal species). In addition, the type of cancer investigated, descriptions of hypoxic conditions (such as oxygen concentration, duration, and induction method), the role of EVs (including the types studied and their role in cancer under hypoxia), ncRNAs investigated (e.g., miRNAs and circRNAs), and their regulation under hypoxic conditions, as well as key results related to the impact of hypoxia on cancer progression, therapeutic implications, and diagnostic applications, were noted. In cases where essential data were incomplete or unavailable, the study authors were contacted to provide the missing information.

### 2.6. Risk of Bias in Individual Studies

To analyze the risk of bias in each article included in this review, the Systematic Review Centre for Laboratory Animal Experimentation risk of bias tool was used to evaluate potential bias in animal studies [28]. This tool is specifically recommended for assessing bias in randomized trials, as included in Cochrane Reviews, but has been adapted to address particular forms of bias unique to animal intervention studies.

### 2.7. Characteristics of Studies

A total of 25 articles were analyzed in this study, including four articles on breast cancer, four on CRC, four on lung cancer, four on hepatocellular carcinoma, three on pancreatic cancer, two on gastric cancer, and one each on oesophageal squamous cell carcinoma, osteosarcoma, bladder cancer, and glioma. These studies explored various types of ncRNAs. Eight articles focused exclusively on circRNAs, whereas 17 assessed more than one type of ncRNA, including circRNAs and miRNAs. All studies utilized cell-treated hypoxia models and included different animal models to explore, understand, and experiment with the molecular mechanisms of ncRNAs involved in tumor growth and metastasis (Table 1).

## 3. Results

The selection process for the studies included in this systematic review began with the identification of 463 publications across the PubMed and Scopus databases. In the initial screening phase, 92 duplicates were removed. Subsequently, a total of 129 articles were excluded as they were not experimental studies. The following eligibility step involved the analysis of titles and abstracts, resulting in the exclusion of 174 studies. Finally, a meticulous full-text evaluation led to the exclusion of 43 additional articles based on specific criteria: 14 did not involve hypoxic conditions, 11 did not utilize animals in the experiments, and 18 employed pharmacological interventions. Upon the conclusion of this selection process, 25 studies were deemed eligible and included in the systematic review. A flow diagram illustrating the selection of studies and the results of the literature search in accordance with PRISMA.

### Risk of Bias

Figure 2 shows the risk of bias in animal studies. Among the 25 studies, 17 were judged to have a high risk of bias, mainly because of the lack of random allocation of animals to different groups and lack of blinding of the outcome investigators. In addition, other details, such as how the issue of missing data was handled and other biases were not adequately addressed in the evaluated studies.

## 4. Discussion

### 4.1. Animal Models in Hypoxia and Cancer Research

The studies discussed in this review describe different techniques for simulating hypoxic conditions that can be applied to either in vitro or in vivo studies. Cells were cultured in a CO_2_ incubator under controlled hypoxic conditions, with oxygen concentrations varying from 0.5–5% to replicate the low-oxygen conditions of the tumor microenvironment. The duration of hypoxic exposure varies depending on the experiment, allowing the study of both acute and chronic effects of low oxygen levels [46,49].

Hypoxia can also be directly induced in animal models. For example, hypoxic cells prepared in vitro are either suspended in saline solutions or in Matrigel and are then injected into animals to create tumor xenografts. Alternatively, they create hypoxic conditions in animals by depriving the organism of oxygen through either localized vascular clamping or the systemic reduction of oxygen levels through the use of hypobaric chambers or specific respiratory interventions [35].

These models are expected to replicate the dynamic and complex interactions of tumor cells with their oxygen-deprived environments, thereby providing insights into tumor progression and therapeutic responses.

In the metastasis model, the caudal vein is a practical choice for tumor cell injection, as it provides easy access to mice and rats, enabling precise and repetitive administration [33,35,39,40,43,49]. This approach closely mimics the natural metastatic process. After entering the systemic circulation, tumor cells follow a path similar to that of metastatic cells in humans, traveling through the bloodstream to colonize distant organs.

The xenotransplantation model, in which human tumor tissue is implanted directly into an animal, offers significant advantages in the study of cancer [36]. Furthermore, it enables a more accurate replication of the original tumor, preserving the cellular architecture and TME, including important tumor components such as stromal cells, blood vessels, and the extracellular matrix. This model better mirrors the tumor characteristics of patients. In addition, xenotransplantation allows the study of tumor progression and invasiveness in the host organism, demonstrating the important aspects of tumor evolution and interaction with the immune system.

Cancer xenotransplantation models are excellent options for testing the responses to personalized therapies, including targeted therapies and immunotherapies. These models allow an accurate assessment of the efficacy of various therapeutic approaches. By examining the features of human cancers, xenograft models enhance the translational relevance of clinical findings. These models are valuable for studying metastases and tumor–host interactions, enabling the evaluation of spontaneous metastatic processes (Figure 3).

### 4.2. Choosing the Animal Model for Experimentation

This review only comprises studies in which the experimental models were mice. This predominance indicates the merits of using mouse models, primarily inbred or genetically modified strains, in cancer research. These mice have excellent genetic uniformity, by which experimental variability can be minimized such that the results are proven consistent and reproducible.

Specific strains of mice, such as BALB/c nude, non-obese diabetic severe combined immunodeficiency (NOD-SCID), and NOD-Prkdc scid IL2rg null (NCG) mice, are ideal candidates for xenograft studies because their immunodeficiency and other immune system features prevent the rejection of implanted human cancer cells or tissues.

Furthermore, mice can be easily subjected to genetic engineering to imitate the biological progression of human cancers, such as tumor growth, metastasis, or even treatment response, making them highly susceptible to human tumor models. Such characteristics, along with extensive access to baseline data and easy practicality in mice owing to their size, low maintenance, and cost, make mice the most suitable model and most frequently used in the studies in this review.

Among the studies included in this review, 15 used BALB/c nude mice experimental model, six used nude mice, two NOD-SCID mice, one NCG mouse, and one BALB NU/NU mouse. BALB/c mice are one of the most widely used inbred models for cancer research.

BALB/C nude mice are BALB/C strain mice with genetic alterations resulting in hairlessness and a lack of T cells, leading to a compromised immune system. They are useful in cancer studies that involve T cell suppression to allow the growth of human tumors without rejection [54].

Nude mice include several variants of immunocompromised mice with a mutation in Foxn1, which causes the absence of T cells and hair. They are widely used in cancer studies, especially for tumor transplantation, owing to their ability to tolerate heterologous tissue grafts [54].

NOD-SCID mice have impairments in both T cells and B cells, making them even more immunocompromised than “nude” mice. They also exhibit impaired natural killer (NK) cells and are more susceptible to spontaneous diabetes. They are ideal for studies that require a virtually absent immune system, allowing for more complex human grafts such as bone marrow transplants and immune response studies [55].

NCG mice were derived from NOD-SCID mice, with an additional deletion of the receptor gamma for interleukin-2 (IL2rg), which affects cells in addition to T and B cells. They offer an extremely immunocompromised environment, being one of the best models for human cell and tissue transplants, ideal for studies of metastatic cancer, and models of human diseases [55].

BALB NU/NU mice are similar to BALB/c nude mice. They are immunodeficient due to a lack of T cells and have genetic characteristics specific to the BALB/c lineage. They are used in tumor transplantation experiments and studies that require the absence of an adaptive immune response to avoid graft rejection [55].

### 4.3. Applicability and Limitations of Animal Models

Hypoxia chambers have been frequently adopted to culture tumor cells at a low oxygen concentration 1% O_2_, to mimic the hypoxic conditions of solid tumors. This facilitates crucial biological activities such as angiogenesis, metabolic reprogramming, and therapeutic resistance mechanisms [29,30,31,32,33]. These animal models help understand the health effects of being in a low-oxygen environment, including the pathway controlled by transcription factors induced by hypoxia that promote tumor formation. However, hypoxia can manifest differently within the TME, exhibiting distinct patterns that influence cancer progression.

Regarding the different types of hypoxia related to cancer progression, chronic and intermittent hypoxia can be highlighted as they affect the organism in distinct ways. Chronic hypoxia is associated with rapid cell proliferation without proper vascularization, resulting in constant oxygen deprivation in regions within the tumor [56,57]. This low-oxygen environment favors the activation of HIFs, which regulate the transcription of genes responsible for cellular adaptation and survival under these conditions [56,57]. In contrast, intermittent hypoxia is characterized by the oscillation of oxygen levels due to variations in blood flow resulting from structural anomalies of tumor blood vessels [58,59]. These oscillations can occur in different periods, promoting more aggressive tumor characteristics, increasing their growth rate, resistance to apoptosis, and a higher metastatic capacity [60].

Additionally, the experimental models were enhanced by variations in hypoxia exposure times, as the cells exhibited different behaviors over time [46,49]. For example, cells were subjected to hypoxic conditions at 1% O_2_ levels for specific periods of 0, 3, 6, 12, 24, and 48 h, allowing for an accurate assessment of the changes caused by hypoxia over time [46]. Similarly, the duration of hypoxia at 1% O_2_ was analyzed over time, reiterating the need to consider the effects of short and prolonged hypoxia on cellular processes such as division, death, and gene activity [49]. The method employed in this study is highly effective for analyzing the duration of hypoxia in human tumors and understanding how it varies significantly in in vitro studies.

The investigation is a crucial effort to recreate situations of chronic intermittent hypoxia in oxygen-deprived rodents, as it was performed in a custom-made box filled with 10% O_2_ and hyperbolically shaped for 12 h per day for six consecutive weeks [35].

This extended protocol investigates factors such as chronic hypoxia that may govern tumor growth, especially metabolic reprogramming, vasculature, and immunological and systemic parameters. However, this model has some limitations, such as the inability to recreate dynamic TMEs containing different degrees of oxygen found in human tumor growth, and likely incompatible immunological responses in mice and humans.

In addition, an investigation using adenomatous polyposis coli-multiple intestinal neoplasia (APC Min/+) mice as a natural model of cancer pathology revealed another approach for assessing hypoxia. This heterozygous mutation (Min/+) causes tumor suppression in mice, leading to spontaneous intestinal adenoma formation and serves as a well-managed model for colorectal cancer (CRC) [35]. The mice were placed under intermittent hypoxia (12 h per day for six weeks), which mimicked a changing TME. This regimen, with intravenous administration of an adenoviral vector targeting the circRNA mmu_circ_0000807, allowed the study of the impact of hypoxia on the proliferation of CRC tumors [35].

The results indicated that hypoxia increased tumor incidence and frequency in mice, with tumor progression observed in the absence of mmu_circ_0000807 [35]. Tumor tissues and serum biomarkers showed elevated levels of mmu_circ_0000807 in response to hypoxic conditions, suggesting its use as an active systemic biomarker and therapeutic target [35]. These tests in animal models are beneficial as they portray dynamic and human-like conditions in the TME, thus increasing our knowledge of the mechanisms controlled by hypoxia in CRC.

Despite these benefits, researchers have improved certain aspects of these models. It remains difficult to replicate the dynamic heterogeneous oxygen gradients typical of human solid tumors and apply these results to the genetic and immunological architectures of heterogeneous patients. Advanced models that incorporate oxygen pacing with in-depth mechanistic exploration can be employed in future studies to identify and characterize novel therapeutic targets of transcendent clinical importance.

### 4.4. Limits and Challenges

The limitations of these experimental methods based on cell culture techniques and animal models are well documented, and the results should not be extrapolated to human health. For example:Tumor heterogeneity: The overall structure and properties of a tumor are determined by the diversity of cell types and gene expression in each tissue. Although cell cultures are clonal, animal models are more complex and do not represent the full biological spectrum of human tumors.Immune responses: The mouse immune system differs from the human immune system, thereby restricting the applicability of the results, particularly regarding hypoxia and its effects on immune modulation.Impairment of the immune system over the total functional capacity in mice was noted in time-limited studies [46,49]. These studies have examples that vary in time to hypoxic extent and its efficiency and efficacy, while prolonged exposure at regular intervals has been advocated [35]. Although helpful in creating a model for recurrent hypoxia, this intermittency may only partially simulate chronic hypoxia, which varies in degree and duration, as observed in the dynamic fluctuations of most human TMEs.Long-term exposure and its consequences: Using 6-week intermittent hypoxic exposure in rats provides critical insights into long-term changes. Such changes are, for example, in tumor vascularization, cellular metabolism, and invasive behavior. One limitation of this model is that it may ignore certain factors that are more critical in a clinical setting, such as the presence of auditory sound gradients or interactions with specific therapies.

### 4.5. Clinical Implications and Therapeutic Perspectives

The analysis of ncRNAs, such as circRNAs and miRNAs, contained in extracellular vesicles (EVs) in hypoxic environments opens new frontiers for oncology. These molecules are not only reflections of tumor biology but also actionable targets, offering dual potential as biomarkers for diagnosis and prognosis and as a basis for new therapeutic strategies. The main results found regarding the biomarker and therapeutic potential of the ncRNAs discussed in this review are summarized in Table 2, Table 3 and Table 4.

### 4.6. Breast Cancer

In breast cancer, hypoxia-regulated ncRNAs are strong indicators of tumor aggressiveness. circPFKFB4, circDENND4C, and circWSB1, along with miR-1287-5p, are associated with a poor prognosis, positioning them as promising non-invasive biomarkers for monitoring disease progression [29,30,31,32].

The therapeutic potential lies in the modulation of these molecular pathways. For example, circPFKFB4, which is induced by HIF1α, has been shown to accelerate breast cancer progression by promoting p27 degradation. Thus, the inhibition of circPFKFB4 has been shown to reduce cell proliferation [29]. Similarly, silencing circDENND4C, which leads to the up-regulation of miR-200b and miR-200c, and circWSB1 resulted in decreased tumor volume and metastatic capacity in experimental models [21,32].

A complementary strategy focuses on the hsa_circ_0001982/miR-1287-5p/MUC19 axis. Hsa_circ_0001982 acts as a sponge for miR-1287-5p, thereby increasing MUC19 expression. Therefore, restoring the levels of the tumor suppressor miR-1287-5p can be achieved through targeted delivery Via EVs or lipid nanoparticles, overcoming the limitations of conventional methods [30].

### 4.7. Colorectal Cancer (CRC)

For CRC, circEIF3K, circ-133, and mmu_circ_0000807 are linked to immune evasion and increased invasiveness, functioning as reliable indicators for early diagnosis and metastatic risk assessment [33,34,35]. Hypoxia-induced exosomal circRNA-133, for instance, promotes metastasis by activating the GEF-H1/RhoA axis. In another mechanism, exosomal circEIF3K promotes CRC progression Via the miR-214/PD-L1 axis, highlighting a pathway for immune evasion.

The dysregulation of miR-133a and miR-214 also correlates with advanced stages and poor outcomes [33,34]. Therapeutically, modulating these axes is promising. It is possible to neutralize the effects of circRNA-133 through the delivery of miR-133a Via EVs [33]. Similarly, restoring miR-214 levels can help reverse tumor progression [34].

Furthermore, inhibiting the expression of mmu_circ_0000807 may restrain metastatic ability. Another identified pathway involves circINSIG1, which reprograms cholesterol metabolism; silencing circINSIG1 has been shown to reduce tumor growth, proliferation, and lipid signaling in CRC, suggesting it as another valuable therapeutic target [34,35,36].

### 4.8. Lung Cancer

In lung cancer, circulating exosomes carrying circPLEKHM1, circ_0007386, and circ_0001875 serve as biomarkers of metastatic potential, allowing for less invasive monitoring of tumor burden [38,39,40]. The identified molecular pathways provide several therapeutic targets. For instance, circ_0000376 was shown to promote tumor growth by regulating the miR-1182/NOVA2 axis, positioning miR-1182 as a marker of good prognosis [37].

In another pathway, circ_0007386 promotes cell proliferation through the miR-383-5p/CIRBP axis. This circRNA was found to be regulated under hypoxia by the YAP1-EIF4A3 interaction [38]. Furthermore, exosomal circPLEKHM1 was identified as an important regulator of cellular communication, driving metastasis and M2 macrophage polarization in the tumor microenvironment [39].

Therapeutic approaches focus on inhibiting these metastasis-promoting circRNAs, such as circ_0007386 and circ_0001875, using siRNAs or blockers delivered by EV-based systems [38,40]. On the other hand, restoring the suppressive effect of miR-1182 and blocking the pro-growth circ_0000376 can inhibit tumor progression [37].

### 4.9. Hepatocellular Carcinoma (HCC)

In HCC, circulating circRNAs are linked to metabolic changes and immune evasion. For example, the hypoxia-associated circPRDM4 promotes immune evasion by reducing CD8+ T cell infiltration and increasing the expression of CD274 (PD-L1) [41]. Another circRNA, circMAT2B, is an identifier of advanced disease that promotes glycolysis and malignancy by regulating the miR-338-3p/PKM2 axis [43].

Other miRNAs and their regulatory axes also interfere with tumor dynamics. Hypoxia-induced hsa_circ_0008450 accelerates HCC progression Via the miR-431/AKAP1 axis [42]. Additionally, under hypoxia, loss of the androgen receptor can promote invasion by activating the circ-LNPEP/miR-532-3p/RAB9A signal pathway, where circ-LNPEP acts as a sponge for miR-532-3p [44].

Therapeutic strategies can target these pathways. Reducing circMAT2B expression, for instance, has been proposed as a strategy. Blocking the interaction of circ_0008450 with miR-431 inhibited cell invasion, and the reintroduction of miR-532-3p Via modified EVs emerges as an effective therapeutic modality to reduce tumor aggressiveness [42].

### 4.10. Gastric Cancer

In gastric cancer, circC6orf132 and circSLAMF6 contribute to tumor adaptation in hypoxic environments by modulating key processes such as glycolysis and cell proliferation, which indicates more aggressive tumors with a higher risk of metastasis [45,46]. CircC6orf132 facilitates proliferation, migration, and glycolysis by acting on the miR-873-5p/PRKAA1 axis [45]. Similarly, circSLAMF6 was shown to regulate cell glycolysis, migration, and invasion Via the miR-204-5p/MYH9 axis [46].

The miRNAs miR-873-5p and miR-204-5p are also dysregulated, affecting survival and therapy resistance, indicating more aggressive tumors with a higher risk of metastasis. The therapeutic potential lies in developing EV-based therapies to modulate these ncRNAs, aiming to reduce cell proliferation and motility, thereby decreasing metastatic potential [45,46].

### 4.11. Pancreatic Cancer

In Pancreatic Cancer circPDK1 and circZNF91 act as prognostic indicators; high expression of circPDK1, which promotes glycolysis by modulating the miR-628-3p/BPTF axis, is associated with a poor prognosis [47,48]. Hypoxic exosomal circZNF91 can predict resistance to gemcitabine, a function it performs by enhancing glycolysis Via the circZNF91/miR-23b-3p axis [48]. Furthermore, circATG7 was found to facilitate autophagy and promote cancer progression through the miR-766-5p/ATG7 and HUR/ATG7 axes [49].

Therapeutically, multiple strategies have been proposed based on these findings. To combat chemoresistance, studies suggest that silencing circZNF91 or the exogenous delivery of a miR-23b-3p mimetic Via EVs can sensitize tumor cells to gemcitabine [48]. Furthermore, reducing circPDK1 expression has been shown to decrease metastasis, and modulating the circATG7/miR-766-5p axis can be used to reduce tumor progression [47,49].

### 4.12. Other Malignancies

This pattern repeats across other malignancies, reinforcing the clinical relevance of ncRNAs in EVs:

### 4.13. Esophageal Cancer

In Esophageal Cancer high levels of exosomal circZNF609 promote metastasis and angiogenesis, partly by regulating the circ-ZNF609/miR-150-5p/VEGFA axis and the HuR/ZO-1 axis [50]. Low levels of the tumor suppressor miR-150-5p are linked to aggressive phenotypes. This suggests therapeutic avenues of blocking circZNF609 or reintroducing miR-150-5p Via EVs [50].

### 4.14. Bladder Cancer

In bladder cancer, the upregulation of circELP3 under hypoxic conditions is a key factor in Tumor adaptation and serves as a predictive biomarker for therapy resistance. Consequently, inhibiting circELP3 has been proposed as a therapeutic strategy to improve chemotherapy efficacy [51].

### 4.15. Osteosarcoma

In osteosarcoma, the presence of circCYP51A1 indicates a higher risk of metastasis. Silencing this circRNA was shown to repress cell progression and glycolysis by regulating the circCYP51A1/miR-490-3p/KLF12 axis. This justifies a therapeutic approach of reintroducing miR-490-3p into tumor cells to inhibit invasion [52].

### 4.16. Glioma

In glioma high levels of hypoxic exosomal circ101491 correlate with disease aggressiveness and poor prognosis by promoting progression Via the circ101491/miR-125b-5p/EDN1 axis. These findings point to a dual therapeutic approach: inhibiting circ101491 while simultaneously reintroducing the tumor suppressor miR-125b-5p Via EVs [53].

In summary, ncRNAs transported by EVs in hypoxic contexts are powerful tools that define tumor aggressiveness and offer a non-invasive method for the dynamic monitoring of various types of cancer. The modulation of these ncRNAs opens new avenues for targeted and more precise therapies, holding the key to personalizing clinical treatment in the future.

### 4.17. Limitations and Challenges

However, there are significant challenges. One major hurdle pertains to the standardization of EV isolation and characterization protocols, as these critical factors are essential for ensuring reproducibility and large-scale applications. In addition, most ncRNAs are functionally redundant, making it difficult to determine their specificity in this complex TME scenario and the relevance of their clinical applicability. This necessitates elaborate studies of ncRNAs to validate their significance in various clinical conditions.

Although animal models are indispensable for preclinical research, they do not fully predict human outcomes. However, they provide valuable insights at the preclinical level and lay the groundwork for understanding how EVs and ncRNAs function in tumor biology and therapeutic applications. Differences in physiology, immune responses, and tumor progression should be recognized because they strongly influence results, which may differ between experiments. Nevertheless, animal testing will still provide meaningful guidance for initial experiments, hypothesis testing, and treatment strategy optimization, which may then proceed to human trials.

### 4.18. Future Perspectives

The integration of extracellular vesicles (EVs) and non-coding RNAs (ncRNAs) into clinical practice holds the potential to revolutionize cancer management. Promising strategies involve the early detection of biomarkers in body fluids Via liquid biopsies, alongside functional ncRNA-based therapeutic approaches. Individualized treatment approaches directed by the specific molecular profiles of the EVs in question are likely to change the therapeutic landscape of cancer, particularly in cases of hypoxia, where the present therapeutic approaches are limited.

An exciting approach already undertaken in some studies is the establishment of cancer in animal models, followed by treatment with EVs to ascertain their efficacy in tumor reduction [54,55,56]. This experimental model allowed us to assess the effects of these EVs on tumor size and their inhibitory effects on circRNAs and miRNAs involved in tumor growth and metastasis. Additionally, it may disrupt critical metabolic pathways such as glycolysis. Therefore, it offers an excellent opportunity for testing new therapies promising to improve patient prognosis through modulation of these ncRNAs, which may contribute to poor patient prognosis.

## 5. Conclusions

This systematic review aimed to evaluate the effects of cellular hypoxia in different types of cancer using exclusively animal models, highlighting the regulatory role of microRNAs and circular RNAs in tumor development. The use of animal models to address this issue provides the most valuable information, filling the gap in the information provided by in vitro models. This will allow us to better understand the mechanisms related to cancer and advance the development of new therapeutic strategies.

Understanding the interactions between these molecules and the perspective of EVs as carriers of ncRNAs could help establish novel ways to advance therapeutic approaches and personalized treatments. These findings call for additional studies to validate these biomarkers in terms of their clinical relevance, thus paving the way for more effective and personalized cancer management in hypoxic environments.

Current data collection involving animal models helps to lay out critical foundations of knowledge; nevertheless, clinical trials are required to evaluate the efficacy and safety of these strategies when applied to human patients. Well-designed clinical studies are necessary to transfer this evidence into practice and secure its possible benefits for treating cancer within realistic contexts.

## Figures and Tables

**Figure 1 biomedicines-13-02796-f001:**
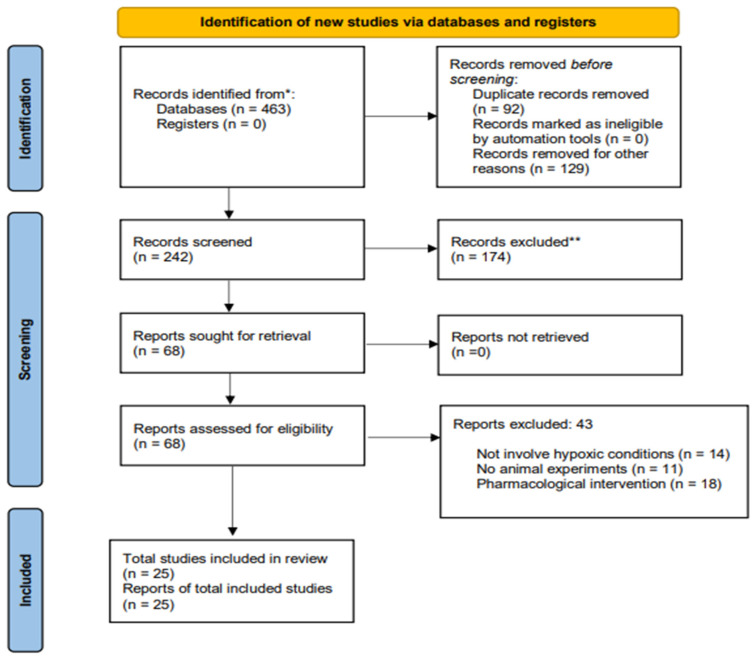
Flow diagram of the current systematic review conducted according to the Preferred Reporting Items for Systematic Reviews and Meta-analysis (PRISMA) guidelines. * Data sources: PubMed and Scopus; search conducted until February 2025. ** Records excluded during initial screening by title and abstract (*n* = 174) for not meeting the pre-defined inclusion criteria for the research question.

**Figure 2 biomedicines-13-02796-f002:**
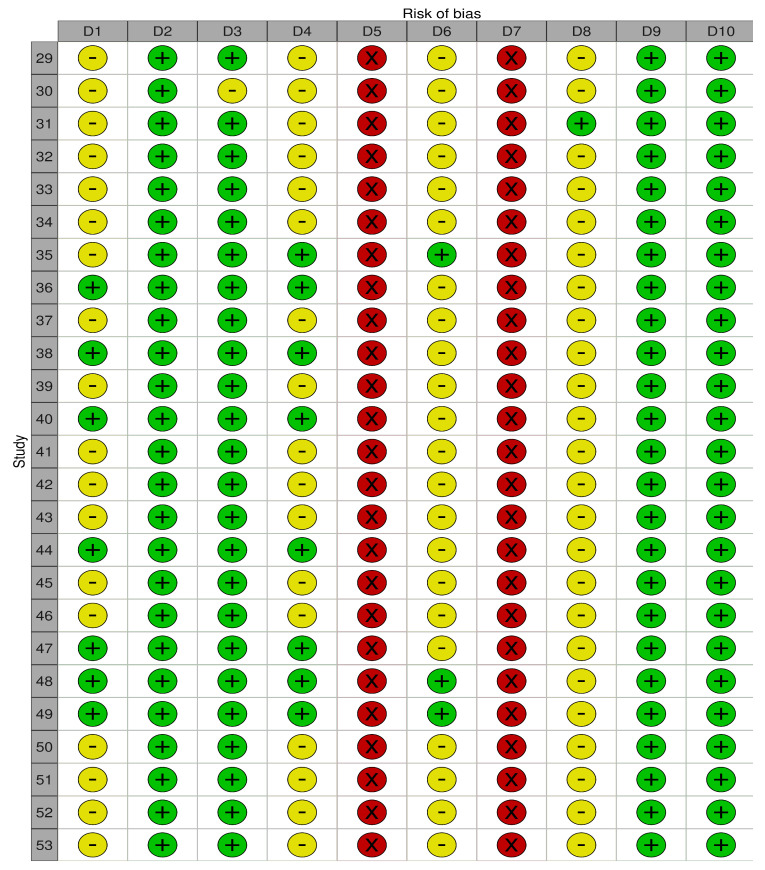
Systematic Review Center for Laboratory Animal Experimentation (SYRCLE) risk of bias summary: the reviewers’ assessments of the risk of bias for each item in every animal study included. D1: Was the allocation sequence adequately generated and applied? D2: Were the groups similar at baseline or were they adjusted for confounders in the analysis? D3: Was the allocation to the different groups adequately concealed during? D4: Were the animals randomly housed during the experiment? D5: Were the caregivers and/or investigators blinded from knowledge which intervention each animal received during the experiment? D6: Were animals selected at random for outcome assessment? D7: Was the outcome assessor blinded? D8: Was incomplete outcome data adequately addressed? D9: Are reports of the study free of selective outcome reporting? D10: Are there other problems that could result in high risk of bias? Green circle with a plus sign (+): Indicates Low Risk of Bias. Yellow circle with a minus sign (−): Indicates Unclear Risk of Bias (Uncertainty). Red circle with an ‘X’: Indicates High Risk of Bias.

**Figure 3 biomedicines-13-02796-f003:**
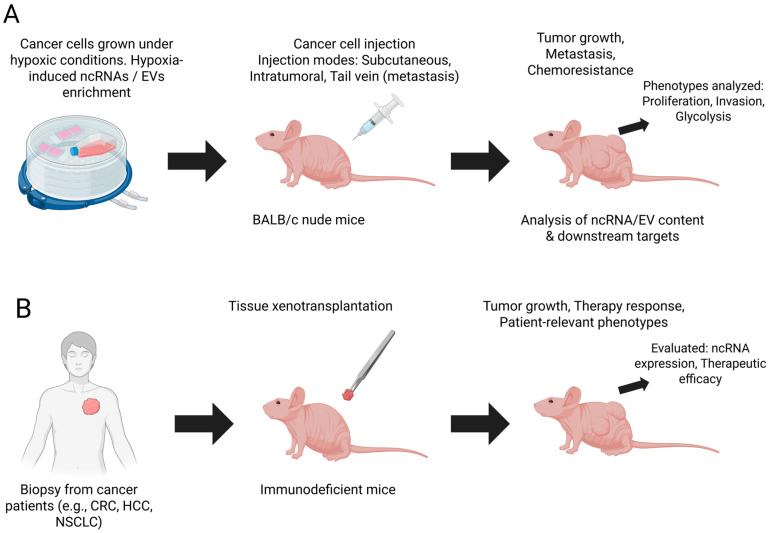
Conceptual overview of the main preclinical animal models used in the reviewed studies. (**A**) The Cell Line-Derived Xenograft (CDX) model. Cancer cell lines are first cultured under hypoxic conditions to study changes in ncRNA expression and EV cargo. These cells are then injected into immunodeficient mice (e.g., BALB/c nude) using various methods (such as subcutaneous injection for tumor growth or tail vein injection for metastasis models). This model is used to evaluate phenotypes such as tumor progression, glycolysis, and invasion. (**B**) The Patient-Derived Xenograft (PDX) model. Tumor tissue is extracted directly from a human patient and implanted into a severely immunodeficient mouse (e.g., NOD/SCID). This model better retains the heterogeneity of the original human tumor and is used to evaluate tumor progression, the hypoxia-ncRNA axis, and therapeutic responses in a more clinically relevant context. Created in BioRender. Afonso, J. (2025) https://BioRender.com/hdd067d (accessed on 11 November 2025).

**Table 1 biomedicines-13-02796-t001:** Main features of each study.

Reference	Cancer Type	Non-Coding RNAs	Hypoxia Conditions	EV Type Studied	Animal Model	Outcomes	Principal Findings
[29]	Breast cancer (BC)	CircPFKFB4	1% O_2_	N/A	BALB/c nude mice (4–6 weeks), injected subcutaneously with MCF-7 cells	Tumor growth	HIF1α-induced circPFKFB4 accelerates breast cancer progression by promoting p27 degradation and leading to a worse prognosis.
[30]	Breast cancer (BC)	circ_0001982, miR-1287-5p, MUC19	Hypoxia chamber with 1% O_2_	N/A	MDA-MB-231 cells or control cells (MCF-10A) expressing sh-circ_0001982 or sh-NC were injected into female nude mice	Tumor growth, metastasis and glycolysis	Knockdown of hsa_circ_0001982 significantly reduced tumor volume and weight. MUC19 expression decreased, while miR-1287-5p expression increased, suggesting that hsa_circ_0001982 regulates MUC19 via sponging for miR-1287-5p.
[31]	Breast cancer (BC)	circDENND4C	Hypoxia chamber with 1% O_2_	N/A	Female BALB/c nude mice were injected subcutaneously with stably transfected MDA-MB-453 cells (5 × 10^6^), termed as sh-circ or sh-NC group	Inhibition of tumor growth and metastasis	Knockdown of circDENND4C significantly reduced tumor volume and weight. The expression of circDENND4C was decreased by 48%, while the expressions of miR-200b and miR-200c were increased by 3.12- and 2.56-fold, respectively. The expression of HIF1A was also reduced in the sh-circ group.
[32]	Breast cancer (BC)	circWSB1	1% O_2_ in tri-gas incubator	N/A	MCF-7 cells (1 × 10^7^) were subcutaneously inoculated into the dorsal flanks of the randomly grouped nude mice	Tumor growth	Overexpression of circWSB1 increases tumor growth, while its suppression reduces growth and improves survival in mice
[33]	Colorectal cancer (CRC)	circRNA-133, miR-133a	1% O_2_ in cell cultures	Exosomes	BALB/c-nude mice with xenografts of HCT116 cells	Tumor growth and metastasis	Exosomes with high circ-133 expression increased tumor volume and weight; increased CTCs; decreased E-cadherin in the membrane; activation of the circ-133/GEF-H1/RhoA axis promoting metastasis; GEF-H1 expression modulated in relation to circ-133.
[34]	Colorectal cancer (CRC)	miR-214, circEIF3K	94% N_2_, 5% CO_2_, 1% O_2_	Exosomes	NOD-SCID mice injected subcutaneously with HCT116 cells	Increased tumor volume and weight	Tumors treated with sh-NC (circEIF3K) exosomes were larger than those treated with sh-circEIF3K. High levels of circEIF3K were associated with advanced stages of the disease and lower survival rates in patients. circEIF3K regulates the miR-214/PD-L1 axis.
[35]	Colorectal cancer (CRC)	Hsa_circ_0000826, mmu_circ_0000807	Hypoxia chamber with 1% O_2_; 10% O_2_ for animals for 12 h per day	N/A	Nude mouse model with SW620 and HCT116 cells.	Tumor growth and metastasis	Hypoxia increased the expression of mmu_circ_0000807 and hsa_circ_0000826, resulting in increased tumorigenesis and metastasis in colorectal cancer. Mice in hypoxia had more tumors and the expression of mmu_circ_0000807 in tumor tissues increased. Inhibition of mmu_circ_0000807 reduced tumor formation, while hsa_circ_0000826 showed an increase in metastasis capacity.
[36]	Colorectal cancer (CRC)	circINSIG1	1% O_2_ and 100 μM CoCl_2_	N/A	DLD1 cells, were injected into the cecal wall of 6-week-old NOD-SCID mice. Tumor tissues from colorectal cancer patients were implanted into NOD-SCID mice	Tumor growth and metastasis	Silencing of circINSIG1 reduces tumor growth, proliferation, and lipid signaling pathway in CRC, suggesting its potential as a therapeutic target.
[37]	Lung cancer (NSCLC)	circ_0000376, miR-1182	1% O_2_, 5% CO_2_ and 94% N_2_	N/A	Male BALB/c mice, divided into two groups (*n* = 6), with transfected H522 cells injected subcutaneously.	Tumor growth	Silencing of circ_0000376 inhibited tumor growth by regulating the miR-1182/NOVA2 axis, highlighting it as a possible therapeutic target.
[38]	Lung cancer (NSCLC)	circ_0007386, CIRBP, YAP1, EIF4A3	Incubation at 37 °C in a hypoxemic chamber with 1% O_2_ and 5% CO_2_	N/A	Female nude mice (BALB/c), 4 weeks old, were subcutaneously injected with 6 × 10^6^ transfected cells.	Tumor growth	circ_0007386 promotes cell proliferation and tumor growth through the miR-383-5p/CIRBP axis, being regulated under hypoxia by the YAP1-EIF4A3 interaction.
[39]	Lung cancer (NSCLC)	circPLEKHM1	Incubation at 37 °C under 1% O_2_	Exosomes	Injection of cancer cells via intratibial or left ventricular route in BALB/c nude mice	Tumor growth and metastasis	circPLEKHM1-ASO treatment significantly reduced NSCLC metastasis (lung and bone), increased mouse survival, and decreased bone destruction and M2 macrophage polarization. CircPLEKHM1 was identified as an important regulator of cellular communication under hypoxia in the TME.
[40]	Lung cancer (NSCLC)	circ-0001875	1% O_2_ in hypoxemic chamber	N/A	Subcutaneous Tumor Model: subcutaneous injection of transfected cells into BALB/c nude mice. Metastasis Studies: injection of transfected cells into the tail vein of mice to monitor lung and liver metastases.	Tumor growth and metastasis	circ-0001875 promotes tumor growth and metastasis in NSCLC; overexpression increases proliferation and metastasis, while inhibition reduces them.
[41]	Hepatocarcinoma (HCC)	circPRDM4	1% O_2_	N/A	NCG mice injected subcutaneously with HCC cells	Tumor growth and immune control	circPRDM4 promotes tumor growth and facilitates immune evasion in hepatocellular cancer models by reducing CD8+ T cell infiltration and increasing the expression of CD274 (PD-L1), a protein linked to immunosuppression.
[42]	Hepatocarcinoma (HCC)	circ_0008450, miR-431	94% N_2_, 5% CO_2_, 1% O_2_ for 48 h	N/A	Huh7 cells containing sh-circ_0008450 (for silencing of circ_0008450) or control (sh-NC) were injected subcutaneously into BALB/c nude mice.	Tumor growth	Silencing of circ_0008450 inhibited tumor growth and modulated the expression of miR-431 and AKAP1.
[43]	Hepatocarcinoma (HCC)	circMAT2B, miR-338-3p	Incubation in a hypoxemic chamber with 1% O_2_	N/A	HCC cells were subcutaneously injected into the flanks of female BALB/c-nude mice.	Tumor growth and metastasis	circMAT2B increases glucose uptake, tumor growth and metastasis in HCC by regulating the miR-338-3p/PKM2 axis. Overexpression of circMAT2B is associated with poor prognosis in HCC patients.
[44]	Hepatocarcinoma (HCC)	miR-532-3p, circ-LNPEP	1% O_2_, 5% CO_2_, 94% N_2_	N/A	BALB/c nude mice injected with SK-HEP-1 luciferase cells	Tumor growth and metastasis	AR overexpression slows tumor growth and metastasis, an effect reversed by the presence of circ-LNPEP. RAB9A expression is increased in the presence of circ-LNPEP, which acts as a sponge for miR-532-3p, promoting cell invasion in HCC.
[45]	Gastric cancer (GC)	circC6orf132, miR-873-5p	1% O_2_	N/A	BALB/c nude mice, injected sh-circC6orf132 or sh-NC transfected HGC-27 cells	Tumor growth and tumor inhibition	Silencing of circC6orf132 reduced tumor growth, glycolysis (with decreased GLUT1, HK2, lactate production and glucose uptake) and the expression of proliferation-related proteins (Ki-67, PCNA). There was an increase in the expression of miR-873-5p and a decrease in PRKAA1 levels.
[46]	Gastric cancer (GC)	circSLAMF6, miR-204-5p	1% O_2_ for 0, 3, 6, 12, 24, and 48 h	N/A	AGS cells transfected with sh-circSLAMF6 or sh-NC were injected into the male nude mice	Tumor growth	Silencing of circSLAMF6 significantly reduced tumor volume and weight in mice, in addition to decreasing MYH9 expression and increasing miR-204-5p.
[47]	Pancreatic cancer (PC)	circPDK1, miR-628-3p	1% O_2_ for 48 h	Exosomes	BALB/c nude mice with subcutaneous injection of MIA PaCa-2 cells	Tumor growth and metastasis	Hypoxic exosomes increased tumor volume and weight; decreased PCNA and vimentin in IHC; increased E-cadherin. Lower number of lung metastatic nodules with sh1-circPDK1 exosomes. High levels of circPDK1 correlated with worse prognosis in patients.
[48]	Pancreatic cancer (PC)	circZNF91, miR-23b-3p	1% O_2_, 5% CO_2_, and 94% N_2_ at 37 °C	Exosomes	BALB/c male nude mice (4 weeks old), treated with gemcitabine (GEM).	Restoration of gemcitabine sensitivity in xenotransplants.	Hypoxic exosomes reduced tumor response to gemcitabine (GEM). However, intratumoral injection of hypoxic exosomes containing siCircZNF91 or a miR-23b-3p mimetic restored sensitivity to GEM. An increase in circZNF91 expression and a decrease in miR-23b-3p were observed in tumors treated with these exosomes.
[49]	Pancreatic cancer (PC)	circATG7, miR-766-5p	1% O_2_ for various durations (0, 3, 6, 12, 24, and 48 h)	N/A	Cell suspension was injected into the right flank of nude mice	Tumor growth and metastasis	circATG7 promoted pancreatic cancer proliferation and metastasis and facilitated autophagy via miR-766-5p/ATG7 and HUR/ATG7 axes.
[50]	Esophageal squamous cell carcinoma (ESCC)	circ-ZNF609, miR-150-5p	5% CO_2_ with 0.5% O_2_	Exosomes	BALB/c nude mice injected with exosomes and ESCC cells	Tumor growth and metastasis	Exosomal circZNF609 enhances metastasis and angiogenesis in ESCC, contributing to tumor growth and vascular endothelial dysfunction. Overexpression of circZNF609 accelerates tumor growth and reduces ZO-1 expression, indicating endothelial barrier dysfunction.
[51]	Bladder cancer	circELP3	Cultivation under 1% O_2_	N/A	Ten nude male mice (3–4 weeks old) injected with T24 cells for tumor growth monitoring	Tumor growth	Downregulation of circELP3 significantly decreased tumor growth in nude mice (*p* = 0.005) and reduced mean tumor weight (*p* < 0.05). CircELP3 was associated with adaptive response to hypoxia and treatment resistance in bladder cancer.
[52]	Osteosarcoma (OS)	circCYP51A1, miR-490-3p	1% O_2_, 5% CO_2_ and 94% N_2_ for intervals of 0 to 48 h	N/A	MG63 or MNNG/HOS cells transfected with sh-NC or sh-circCYP51A1 were injected into 10 six-week-old male BALB/c nude mice.	Tumor growth	Silencing of circCYP51A1 reduces tumor growth, lung metastasis and expression of miR-490-3p/KLF12 pathway-associated genes in osteosarcoma.
[53]	Glioma	Circ101491, miR-125b-5p	1% O_2_ (hypoxia) and 20% O_2_ (normoxia)	Exosomes	U251S cells were injected subcutaneously into BALB/c-nu/nu mice to generate a tumor model, while U118 cells treated with U251S exosomes were injected into the tail vein to establish a lung metastasis model.	Tumor growth and metastasis	Overexpression of circ101491 increases tumor weight and volume, promotes lung metastasis and is associated with miR-125b-5p inhibition.

Note: N/A: Not Applicable, indicating that extracellular vesicles (EVs) were not the focus of the study or were not specifically mentioned and specified in this work.

**Table 2 biomedicines-13-02796-t002:** Association of Hypoxia-Regulated circRNAs with Tumor Progression and Prognosis in Different Cancer Types.

References	Cancer Type	Animal Model	Non-Coding RNA (circRNA)	Function/Regulation in Hypoxia	Association with Prognosis/Aggressiveness
[29,31,32]	Breast Cancer	BALB/c nude mice (4–6 weeks), injected subcutaneously with MCF-7 cells [29] BALB/c nude mice injected with MDA-MB-453 cells [31] MCF-7 cells (1 × 10^7^) were subcutaneously inoculated into the dorsal flanks of the randomly grouped nude mice [32]	circPFKFB4, circDENND4C, circWSB1	Regulation by HIF1α and other factors in hypoxic environment	Associated with worse prognosis and higher tumor aggressiveness
[33,34,35]	Colorectal Cancer	BALB/c-nude mice with xenografts of HCT116 cells [33] NOD-SCID mice injected subcutaneously with HCT116 cells [34] Nude mouse model with SW620 and HCT116 cells [35]	circEIF3K, circ-133, mmu_circ_0000807	Increased invasiveness and immune evasion	Biomarkers for early diagnosis and metastasis risk monitoring
[38,39,40]	Lung Cancer	Female BALB/c nude mice (subcutaneous injection) [38] Injection of cancer cells via intratibial or left ventricular route in BALB/c nude mice [39] BALB/c nude mice (subcutaneous tumor and tail vein metastasis models) [40]	circPLEKHM1, circ_0007386, circ_0001875	Metastatic potential and tumor aggressiveness	Biomarkers for aggressive tumors and progression; detectable in circulating exosomes
[41,43]	Hepatocellular Carcinoma (HCC)	NCG mice injected subcutaneously with HCC cells [41] HCC cells were subcutaneously injected into the flanks of female BALB/c-nude mice [43].	circMAT2B, circPRDM4	Metabolic changes and immune suppression	Associated with tumor invasion, progression, and immune impairment
[45,46]	Gastric Cancer	BALB/c nude mice, injected sh-circC6orf132 or sh-NC transfected HGC-27 cells [45] AGS cells transfected with sh-circSLAMF6 or sh-NC were Injected into the male nude mice [46]	circC6orf132, circSLAMF6	Modulation of glycolysis and cell proliferation	Indicators of aggressive tumors with higher metastasis risk
[47,48]	Pancreatic Cancer	BALB/c nude mice with subcutaneous injection of MIA PaCa-2 cells [47] BALB/c male nude mice (4 weeks old), treated with gemcitabine (GEM) [48]	circPDK1, circZNF91	Resistance to treatments (gemcitabine)	Prognostic indicators and treatment resistance
[51]	Bladder Cancer	Ten nude male mice (3–4 weeks old) injected with T24 cells for tumor growth monitoring [51]	circELP3	Treatment resistance and adaptation to hypoxia	Indicator of therapeutic resistance and poorer treatment response
[52]	Osteosarcoma	BALB/c nude mice injected with MG63 or MNNG/HOS cells [52]	circCYP51A1	Metastatic potential	Indicator of increased risk of metastasis
[53]	Glioma	BALB/c-nu/nu mice (U251S subcutaneous and U118 tail vein models) [53]	circ101491	Cell proliferation and aggressiveness	Associated with worse tumor aggressiveness and unfavorable prognosis

**Table 3 biomedicines-13-02796-t003:** Hypoxia-Regulated miRNAs and Their Impact on Tumor Progression and Prognosis Across Cancer Types.

References	Cancer Type	Animal Model	Non-Coding RNAs (miRNAs)	Role/Function	Association with Prognosis and Aggressiveness
[30]	Breast Cancer	Female nude mice injected with MDA-MB-231 or MCF-10A cells [30]	miR-1287-5p	Regulates critical pathways, impacts tumor progression under hypoxia	Low levels are linked to tumor aggressiveness and poor prognosis, as it functions as a tumor suppressor.
[33,34]	Colorectal Cancer	BALB/c-nude mice with xenografts of HCT116 cells [33] NOD-SCID mice injected subcutaneously with HCT116 cells [34]	miR-133a, miR-214	Modulate disease progression and immune evasion	Dysregulation correlates with advanced tumor stages and poor outcomes
[37,38]	Lung Cancer	BALB/c mice injected with H522 cells [37] Female BALB/c nude mice (subcutaneous injection) [38]	miR-1182, miR-383-5p	Modulates cell proliferation under hypoxic conditions	miR-1182 is a favorable biomarker for prognosis, correlating with better outcomes
[42,43,44]	Hepatocellular Carcinoma	BALB/c nude mice injected with Huh7 cells [42] HCC cells were subcutaneously injected into the flanks of female BALB/c-nude mice. [43] BALB/c nude mice injected with SK-HEP-1 luciferase cells [44]	miR-431, miR-338-3p, miR-532-3p	Regulate tumor dynamics, metabolism, and immune suppression	High levels of miR-431 and miR-338-3p are associated with poor survival rates
[45,46]	Gastric Cancer	BALB/c nude mice, injected sh-circC6orf132 or sh-NC transfected HGC-27 cells [45] AGS cells transfected with sh-circSLAMF6 or sh-NC were Injected into the male nude mice [46]	miR-873-5p, miR-204-5p	Regulate cell survival and therapy resistance pathways	High expression correlates with therapy resistance and advanced disease
[47,48,49]	Pancreatic Cancer	BALB/c nude mice with subcutaneous injection of MIA PaCa-2 cells [47] BALB/c male nude mice (4 weeks old), treated with gemcitabine (GEM) [48] Cells injected into the right flank of nude mice [49]	miR-628-3p, miR-23b-3p, miR-766-5p	Modulate tumor progression and resistance to therapies	Dysregulation correlates with poor survival rates and aggressive tumor progression
[50]	Esophageal Cancer	BALB/c nude mice injected with exosomes and ESCC cells [50]	miR-150-5p	Critical modulator of tumor progression	Low levels are linked to aggressive phenotypes
[52]	Osteosarcoma	BALB/c nude mice injected with MG63 or MNNG/HOS cells [52]	miR-490-3p	Regulates invasion and metastasis via KLF12 pathways	Downregulation is associated with high metastatic potential and poor outcomes
[53]	Glioma	BALB/c-nu/nu mice (U251S subcutaneous and U118 tail vein models) [53]	miR-125b-5p	Regulates tumor processes under hypoxia	Low expression correlates with poor prognosis and aggressive tumor characteristics

**Table 4 biomedicines-13-02796-t004:** Potential therapeutic strategies with non-coding RNAs.

References	Cancer Type	Animal Models	circRNAs/miRNAs	Function	Proposed Therapeutic Strategy
[29]	Breast Cancer	BALB/c nude mice (4–6 weeks), injected subcutaneously with MCF-7 cells [29]	circPFKFB4	Increases p27 degradation, promoting tumor aggressiveness	Inhibit circPFKFB4 to reduce cell proliferation; potential delivery of inhibitors via EVs.
[30]	Breast Cancer	Female nude mice injected with MDA-MB-231 or MCF-10A cells [30]	miR-1287-5p	Acts as a tumor suppressor, reducing proliferation and promoting apoptosis	Increase miR-1287-5p levels in tumor cells using EVs or nanoparticles.
[33,34]	Colorectal Cancer	BALB/c-nude mice with xenografts of HCT116 cells [33] NOD-SCID mice injected subcutaneously with HCT116 cells [34]	circEIF3K, circRNA-133	Associated with immune evasion and tumor invasiveness; circRNA-133 regulates miR-133a	Deliver miR-133a via EVs to counteract the pro-tumoral effects of circRNA-133.
[35]	Colorectal Cancer	Nude mouse model with SW620 and HCT116 cells [35]	mmu_circ_0000807	Related to metastatic potential	Block mmu_circ_0000807 expression to reduce metastatic capacity.
[40]	Lung Cancer	BALB/c nude mice (subcutaneous tumor and tail vein metastasis models) [40]	circ_0007386, circ_0001875	Linked to metastasis and tumor aggressiveness	Silence with siRNAs or blockers delivered by extracellular vesicles (EVs).
[37]	Lung Cancer	BALB/c mice injected with H522 cells [37]	miR-1182	Acts as a tumor suppressor, reducing cell proliferation	Introduce exogenous miR-1182 via EVs or lipid nanoparticles.
[43]	Hepatocellular Carcinoma	HCC cells were subcutaneously injected into the flanks of female BALB/c-nude mice [43].	circMAT2B	Regulates tumor metabolism and immune suppression; associated with advanced stages	Reduce circMAT2B expression using CRISPR/Cas9 or specific inhibitors transported by EVs.
[42]	Hepatocellular Carcinoma	BALB/c nude mice injected with Huh7 cells [42]	circ_0008450	Regulates miR-431, promoting cell survival and tumor invasion	Modulate circ_0008450/miR-431 interaction to inhibit invasion and metastasis.
[44]	Hepatocellular Carcinoma	BALB/c nude mice injected with SK-HEP-1 luciferase cells [44]	miR-532-3p	Related to molecular regulatory processes in HCC	Reintroduce miR-532-3p into tumors via modified EVs.
[45,46]	Gastric Cancer	BALB/c nude mice, injected sh-circC6orf132 or sh-NC transfected HGC-27 cells [45]	circC6orf132, circSLAMF6	Regulate metabolism and cell resistance under hypoxia; associated with proliferation and invasion	Develop therapies to silence these circRNAs, reducing aggressiveness and resistance.
[45,46]	Gastric Cancer	AGS cells transfected with sh-circSLAMF6 or sh-NC were Injected into the male nude mice [46]	miR-873-5p, miR-204-5p	Tumor suppressors, regulating survival and proliferation pathways	Increase levels of these miRNAs to inhibit tumor growth and resistance.
[47,49]	Pancreatic Cancer	BALB/c nude mice with subcutaneous injection of MIA PaCa-2 cells [47]	circPDK1, circZNF91	Associated with poor prognosis and gemcitabine resistance	Deliver target miRNAs like miR-628-3p to reverse resistance; reduce circPDK1 expression.
[49]	Pancreatic Cancer	Cells injected into the right flank of nude mice [49]	miR-766-5p	Modulated by circATG7, related to tumor survival	Modulate miR-766-5p to reduce tumor aggressiveness.
[50]	Esophageal Cancer	BALB/c nude mice injected with exosomes and ESCC cells [50]	circZNF609	Associated with tumor advancement and poor prognosis	Develop therapies to block circZNF609, reducing tumor progression.
[50]	Esophageal Cancer	BALB/c nude mice injected with exosomes and ESCC cells [50]	miR-150-5p	Tumor suppressor, reduces cellular aggressiveness	Introduce miR-150-5p into tumor cells via EVs to limit progression.
[51]	Bladder Cancer	Ten nude male mice (3–4 weeks old) injected with T24 cells for tumor growth monitoring [51]	circELP3	Associated with treatment resistance	Inhibit circELP3 to improve therapeutic efficacy.
[52]	Osteosarcoma	BALB/c nude mice injected with MG63 or MNNG/HOS cells [52]	circCYP51A1	Related to higher metastatic risk	Block circCYP51A1 expression to reduce metastatic risk.
[52]	Osteosarcoma	BALB/c nude mice injected with MG63 or MNNG/HOS cells [52]	miR-490-3p	Regulates invasion and metastasis	Increase miR-490-3p levels in tumors to inhibit metastasis.
[53]	Glioma	BALB/c-nu/nu mice (U251S subcutaneous and U118 tail vein models) [53]	circ101491	Associated with greater aggressiveness and poor prognosis	Develop therapies to inhibit circ101491, reducing tumor progression.
[53]	Glioma	BALB/c-nu/nu mice (U251S subcutaneous and U118 tail vein models) [53]	miR-125b-5p	Tumor suppressor, reduces cellular aggressiveness	Reintroduce miR-125b-5p to limit tumor proliferation.

## Data Availability

The data presented in this study will be made available upon request to the corresponding author.

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
