# Peer review of "Hypoxia-Induced Extracellular Vesicles and Non-Coding RNAs in Cancer: A Systematic Review of Tumor Dynamics and Therapeutic Implications in Preclinical Animal Models"

_biomedicines, 2025, doi:10.3390/biomedicines13112796_

Round 1
Reviewer 1 Report
Comments and Suggestions for Authors
The manuscript entitled "Hypoxia-Induced Extracellular Vesicles and Non-Coding RNAs in Cancer: A Systematic Review of Tumor Dynamics and Therapeutic Implications in Preclinical Animal Models" is a good review work of the available literature. However, I have some suggestions for the overall improvement of the manuscript. The comments are listed below;
- The abstract should contain major findings of the literature
- The introduction may sufficiently expand the details about different types of non coding RNAs and their physiological significance
- The molecular mechanism including signaling cascades involved needs to be mentioned. The manipulation of these pathways by different ncRNAs needs to be emphasized, especially while mentioned different stages of cancer.
- Figure 1 quality is low and it needs to be improved
- The figure 2 needs to be improved; text size in the figure needs to be increased to make it readable
- The experimental models in Table 2 to 4 needs to be mentioned.
- There are some typographic errors in the manuscript; corrections can be done with a thorough reading
Author Response
Dear Reviewer 1,
We would like to thank you immensely for your detailed review and constructive comments on the manuscript "Hypoxia-Induced Extracellular Vesicles and Non-Coding RNAs in Cancer: A Systematic Review of Tumor Dynamics and Therapeutic Implications in Preclinical Animal Models". Your suggestions were extremely valuable, and we believe the implemented revisions have significantly improved the quality and clarity of our work.
Below, we detail each of your comments followed by the actions we have taken to address them:
Comment 1: The abstract should contain major findings of the literature.
Response: We thank you for this suggestion. The reviewer is correct in noting that the abstract was focused on general objectives. We have revised the abstract to include the specific major findings from our review. We added sentences that highlight the concrete ncRNAs identified, such as circPFKFB4 in breast cancer and miR-1287-5p, and their consistent association with worse prognosis and therapeutic resistance, as detailed in the results tables.
Comment 2: The introduction may sufficiently expand the details about different types of non coding RNAs and their physiological significance.
Response: We agree that this information is crucial for the reader's understanding. We have expanded the introduction by adding a new paragraph immediately after listing the ncRNA types. This paragraph now details the physiological mechanisms of miRNAs (as post-transcriptional regulators leading to mRNA degradation) and circRNAs (explaining their function as "miRNA sponges").
Comment 3: The molecular mechanism including signaling cascades involved needs to be mentioned. The manipulation of these pathways by different ncRNAs needs to be emphasized, especially while mentioned different stages of cancer.
Response: This was a central observation, and we are grateful for it. We performed an extensive revision of the "Clinical Implications and Therapeutic Perspectives" section . As requested, we rewrote the subsections for each cancer type (e.g., Breast Cancer, Colorectal Cancer, HCC, etc.) to explicitly emphasize the specific signaling cascades and molecular axes identified in the literature (e.g., circ-133/GEF-H1/RhoA axis, miR-214/PD-L1 axis, miR-338-3p/PKM2 axis, etc.). The text now highlights how the manipulation of these axes by ncRNAs affects tumor progression.
Comment 4: Figure 1 quality is low and it needs to be improved.
Response: The reviewer is correct. The image for Figure 1 was of low quality. To correct this, a higher quality image has been put in place of the previous version.
Comment 5: The figure 2 needs to be improved; text size in the figure needs to be increased to make it readable.
Response: We fully agree that Figure 2 was illegible. To resolve this, we have cropped the text section (the description of domains D1-D10) from the image file. The full description for each domain has been added to the figure caption directly within the manuscript text , ensuring full legibility.
Comment 6: The experimental models in Table 2 to 4 needs to be mentioned.
Response: Although our original intention was to avoid redundancy (as the data was in Table 1), we agree with the reviewer that the synthesis tables (Tables 2, 3, and 4) benefit from including this information. We have added a new column titled "Animal Model(s)" to each of these three tables and populated it with the corresponding experimental models, making the summarized results clearer and more complete.
Comment 7: There are some typographic errors in the manuscript; corrections can be done with a thorough reading.
Response: We appreciate you pointing this out. We performed a thorough reading of the entire manuscript and corrected multiple typographic, grammatical, and formatting errors.
Once again, we thank you for your valuable comments, which were fundamental to the improvement of our manuscript.
Reviewer 2 Report
Comments and Suggestions for Authors
This manuscript focuses on an important aspect of tumor progression that warrants further investigation. Both the effects of hypoxia and its relationship with ncRNAs have been addressed in different studies.
In this review, the authors selected studies that report only experimental data from mice, and the data are well presented in tables, allowing for quick visualization. The relevant points are discussed, and the authors discuss and conclude about the relevance of the meaning of ncRNAs in cancer progression. What is the possible contribution of different ncRNAs to the prognosis, diagnosis, and therapy of different types of cancer?
I recommend the manuscript publication .
Author Response
Dear Reviewer 2,
We would like to thank you very much for your time and positive assessment of our manuscript, "Hypoxia-Induced Extracellular Vesicles and Non-Coding RNAs in Cancer: A Systematic Review of Tumor Dynamics and Therapeutic Implications in Preclinical Animal Models".
We are very pleased to hear that you found the topic important and that the data were "well presented in tables, allowing for quick visualization." We also appreciate your recognition that the relevant points were well-discussed.
Regarding your key summary question—"What is the possible contribution of different ncRNAs to the prognosis, diagnosis, and therapy of different types of cancer?"—we are grateful that you highlighted it. This was, indeed, the central objective of our review: to synthesize the evidence from animal models on precisely this threefold potential (prognostic, diagnostic, and therapeutic) of ncRNAs in hypoxic environments.
We sincerely thank you for your recommendation for the manuscript's publication.

Reviewer 3 Report
Comments and Suggestions for Authors
In the review article “Hypoxia-Induced Extracellular Vesicles and Non-Coding RNAs in Cancer: A Systematic Review of Tumor Dynamics and Therapeutic Implications in Preclinical Animal Models”, João Pedro R. Afonso et al., have discussed and summarized how cellular hypoxia (low oxygen conditions) affects cancer development and progression, focusing specifically on animal model studies. The aim of the review is to evaluate how hypoxia impacts different types of cancer in animal models and to understand how microRNAs and circular RNAs regulate tumor development under these conditions. The authors performed the systematic review using PubMed/MEDLINE and Scopus databases. Out of 171 identified studies, 25 met the inclusion criteria. Animal models provide detailed insight into how cancers behave under hypoxic conditions and in some cancers, EVs released by cells can promote tumor growth and metastasis. The review demonstrates the robustness of animal models for studying the molecular mechanisms of cancer and for developing new therapeutic strategies.
Major comments:
- The brief introduction of the types of the EVs if missing: In the introductory part of the review article the authors should explain briefly regarding the different types of EVs as summarized in the literature (PMID: 36737375, PMID: 33304471). To give a brief context to the types of EVs would provide with the readers to have the starting point and then dive in the main theme of the review i.e., how cellular hypoxia affects cancer development and progression.
- Table 1. Main features of each study: I would highly encourage the authors to add a separate section of the types of the EVs focused on that study. To add a column describing the type of the EVs that is involved in that study would provide more clarity for the readers and would be convenient to summarize the research findings.
- Figure 3: The figure looks too simplistic with no significant information for the readers. I would highly encourage the authors to add more information in the figures. This should be in accordance to table 1. The figure should include different mouse models and the mode of injection done for them, origin of the cancer cells and the development of the mouse model, and the EVs secreted from that particular cancer and the miRNAs associated, and lastly the fate/phenotype of the model and what value does it add in studying the scientific question of how hypoxia impacts different types of cancer.
Author Response
Dear Reviewer 3,
We would like to express our sincere gratitude for your insightful comments, which identified important areas for refinement of our manuscript. Your suggestions have helped us to significantly improve the clarity and methodological rigor of our review.
Below are our detailed responses to your concerns:
Comment 1: The brief introduction of the types of the EVs is missing: In the introductory part of the review article the authors should briefly explain the different types of EVs as summarized in the literature (PMID: 36737375, PMID: 33304471). Providing a brief context on the types of EVs would give readers a starting point before delving into the main theme of the review...
Response: We appreciate this important suggestion. The reviewer is correct in noting that an introduction to the different types of EVs would enrich the context for the readers. Following the suggestion, we have added a new paragraph to the Introduction that briefly defines the main types of EVs (Exosomes, Microvesicles, and Apoptotic Bodies) based on their biogenesis and size, and cited the recommended literature.
Comment 2: Table 1. Main features of each study: I would highly encourage the authors to add a separate section on the types of EVs focused on in each study. Adding a column describing the type of EVs involved in each study would provide more clarity for the readers and would be convenient for summarizing the research findings.
Response: This is an excellent suggestion that identified a central inconsistency in our presentation. To increase clarity and address the reviewer's request, the title and inclusion criteria have been rewritten for greater clarity.
The original intention of the review was to evaluate the effect of hypoxia on ncRNAs in animal models (a criterion that all 25 articles meet) and, within that group, to analyze the subset of studies that also investigated EVs as carriers. Our original criteria did not accurately reflect this. To correct this inconsistency and align the scope:
- The Eligibility Criteria were rewritten to clarify that the primary inclusion criterion was (Hypoxia + ncRNAs + Animal Model), and that studies that also investigated EVs were a subset of interest.
- As requested, we added the new column 'EV Type Studied' to Table 1. This column now clearly identifies which studies focused on EVs (e.g., 'Exosomes') and which focused on the ncRNA mechanism itself (marked as 'N/A').
We believe this approach aligns the methodology with the data presented and provides the requested clarity.
Comment 3: Figure 3: The figure looks too simplistic with no significant information for the readers. I would highly encourage the authors to add more information in the figures. This should be in accordance with table 1. The figure should include different mouse models and the mode of injection done for them, origin of the cancer cells and the development of the mouse model, and the EVs secreted from that particular cancer and the miRNAs associated, and lastly the fate/phenotype of the model...
Response: We appreciate the suggestion. We would like to clarify that the intention of Figure 3 is to be a conceptual diagram to illustrate the two animal model methodologies discussed (cell injection vs. tissue xenotransplantation), and not a summary of results.
We agree that the original figure was too simplistic. However, we believe that compiling all the details of the 25 studies (models, cells, ncRNAs, EVs, and phenotypes) into a single graph would make it excessively dense and illegible. We believe that Tables 1, 2, 3, and 4 are the most appropriate place to present this detailed data.
To address the reviewer's request and improve the figure, we have redesigned it to be more methodologically informative:
We added more details to the labels (such as examples of injection modes, e.g., 'Subcutaneous injection' or 'Tail vein injection'). We specified the types of models (e.g., 'Cell Line-Derived Xenograft (CDX) model' and 'Patient-Derived Xenograft (PDX) model') and the outcomes analyzed (e.g., 'Evaluate tumor growth, metastasis...'). The caption for Figure 3 was completely rewritten to be more descriptive and explain what each conceptual model represents.
We believe that the new Figure 3, together with its detailed caption, now provides a much richer methodological context.
Once again, thank you for your constructive comments.

Round 2
Reviewer 1 Report
Comments and Suggestions for Authors
The authors have addressed the comments raised by the reviewer, and I am happy with the response. The manuscript can be considered for further processing.